# Sea Ice Climate Normals for Seasonal Ice Monitoring of Arctic and Sub-Regions

**Ge Peng** [1,*] , **Anthony Arguez** [2] , **Walter N. Meier** [3] , **Freja Vamborg** [4] , **Jake Crouch** [2] **and Philip Jones** [5]

1   Cooperative Institute for Climate and Satellites—North Carolina (CICS-NC) at NOAA's National Centers for Environmental Information (NCEI), North Carolina State University, Asheville, NC 28801, USA
2   NOAA/National Centers for Environmental Information (NCEI), Asheville, NC 28801, USA
3   National Snow and Ice Data Center (NSIDC), Cooperative Institute for Research in Environmental Sciences (CIRES), University of Colorado, Boulder, CO 80309, USA
4   The European Centre for Medium-Range Weather Forecasts (ECMWF), Reading RG2 9AX, UK
5   Riverside Technology, Inc., Asheville, NC 28801, USA
*   Correspondence: gpeng@ncsu.edu; Tel.: +1-828-257-3009

**Abstract:** The climate normal, that is, the latest three full-decade average, of Arctic sea ice parameters is useful for baselining the sea ice state. A baseline ice state on both regional and local scales is important for monitoring how the current regional and local states depart from their normal to understand the vulnerability of marine and sea ice-based ecosystems to the changing climate conditions. Combined with up-to-date observations and reliable projections, normals are essential to business strategic planning, climate adaptation and risk mitigation. In this paper, monthly and annual climate normals of sea ice parameters (concentration, area, and extent) of the whole Arctic Ocean and 15 regional divisions are derived for the period of 1981–2010 using monthly satellite sea ice concentration estimates from a climate data record (CDR) produced by NOAA and the National Snow and Ice Data Center (NSIDC). Basic descriptions and characteristics of the normals are provided. Empirical Orthogonal Function (EOF) analysis has been utilized to describe spatial modes of sea ice concentration variability and how the corresponding principal components change over time. To provide users with basic information on data product accuracy and uncertainty, the climate normal values of Arctic sea ice extents (SIE) are compared with that of other products, including a product from NSIDC and two products from the Copernicus Climate Change Service (C3S). The SIE differences between different products are in the range of 2.3–4.5% of the CDR SIE mean. Additionally, data uncertainty estimates are represented by using the range (the difference between the maximum and minimum), standard deviation, 10th and 90th percentiles, and the first, second, and third quartile distribution of all monthly values, a distinct feature of these sea ice normal products.

**Dataset:** The dataset DOI: https://doi.org/10.25921/TRXE-M983.

**Dataset License:** License under which the dataset is made available (CC-BY 4.0).

**Keywords:** climate normal; Arctic; sea ice; decadal trend; variability; climate data record; EOF; NSIDC; Copernicus; NOAA

---

## 1. Summary

Arctic sea ice coverage has been undergoing rapid depletion since satellite-based measurements became available in the late 1970s, especially the summer ice coverage, e.g., [1–3]. Sea ice decline

is most pronounced in the coastal areas such as the Laptev, East Siberian, Chukchi, and Beaufort seas [3,4]. Arctic summer circulation may contribute to regional sea ice anomalies [5]. Spatial sea ice variability may lead to a large spread in climate model sea ice area projections and therefore induces high uncertainty on regional scales [6]. Superimposed on this strong downward trend is the pronounced inter-annual variability and those two characteristics are essentially not correlated [7].

Long-term averages on both regional and local scales are important for monitoring how the current regional and local states depart from their normal conditions to understand the variability and therefore potential vulnerability of marine and sea ice-based ecosystems and habitats. Combined with up-to-date observations and reliable climate projections, normals are essential to business strategic planning, climate adaptation and risk mitigation.

To help establish baselines for the climate state, monthly and annual climate normals of sea ice parameters (concentration, area, and extent) of the whole Arctic Ocean and 15 regional divisions, are derived using monthly data files from the NOAA and National Snow and Ice Data Center (NSIDC) passive microwave sea ice concentration climate data record (CDR) dataset [8]. The method of using the arithmetic average over the last three complete decades (1981–2010) as defined by the World Meteorological Organization [9] is adapted. Basic descriptions and characteristics of the normals are provided. The spatial modes of sea ice concentration variability within the time period are examined utilizing Empirical Orthogonal Function (EOF) analysis. To provide users with basic quality and uncertainty information on the data product, the climate normal values of Arctic sea ice extents (SIE) from the CDR are compared with those of another product from NSIDC and two products from the European Union's Copernicus Climate Change Service (C3S). The differences between them range from 0.285 to 0.55 ($10^6$ km$^2$), which is about 2.3–4.5% of the Arctic SIE mean for the period of 1981–2010. In addition, data uncertainty is represented by using the range (the difference between the maximum and minimum), standard deviation, and 10th and 90th percentiles and quartile distribution of all monthly values.

## 2. Data Description

### 2.1. Input Data Description

Monthly sea ice concentration fields from the NOAA/NSIDC CDR are utilized to derive the climate normals presented here. The CDR is a long-term, consistent, satellite-based passive microwave record of sea ice concentration. This sea ice concentration product leverages two well-established concentration algorithms, the NASA Team (NT) and Bootstrap (BT). Both algorithms were developed and produced by the NASA Goddard Space Flight Center (GSFC) [8]. Description and verification of the CDR dataset can be found in [10] and [11], respectively. The CDR data files used in this study are from the version v03r01 [8].

The CDR data files include two primary sea ice concentration (SIC) parameters: the CDR concentration and similar Goddard Merged concentration. We will hereafter refer to them as CDR SIC and GSFC$_m$ SIC, respectively. Two additional GSFC-derived SIC fields are also included in each CDR data file: GSFC-derived NT and BT. As the current CDR SIC only spans the years from 1987 to 2017, which is short of the three full decades required for a WMO-defined climate normal, GSFC$_m$ SIC will be used instead. GSFC$_m$ SIC is derived using the same processing algorithm as that for CDR SIC, but using the manually quality-controlled GSFC-derived NT and BT sea ice concentrations as input data sources [8]. Manual quality-control means sea ice concentration values may be modified manually at the cell level by examining concentration distributions. The approach is subjective and not reproducible. Furthermore, there is currently no quality flag to indicate if the value has been modified manually. All the monthly SIC fields are on the NSIDC polar stereographic grid with nominal 25 km × 25 km grid cells [10].

The monthly Arctic GSFC$_m$ SIE climate normals are compared with that derived from three other sea ice products. The first comparison product is the NSIDC Sea Ice Index (SII) monthly sea ice extent

values [12]. The SII monthly sea ice coverage values are computed from the daily SIC values using the NT algorithm product archived at NSIDC [13], which is also the source data for the monthly GSFC-derived NT field in the CDR data files [10].

The two other climate normal comparison products are computed from the ERA-Interim and ERA5 dataset, provided through C3S. ERA-Interim and ERA5 are the global reanalyses produced by the European Centre for Medium-Range Weather Forecasts (ECMWF), e.g., [14,15]. Neither ERA-Interim nor ERA5 contains prognostic parts for sea ice and as such conditions for sea-ice cover (SIC) need to be prescribed from other products. The source sea ice data are from the Met Office Hadley Centre sea ice products, e.g., [16] and the EUMETSAT OSI-SAF satellite product [17]. The sea ice cover in ERA-Interim and ERA5 differs from the original sea ice products for several reasons, including re-gridding and removal of sea ice based on physical consistency checks with temperature [15]. For the calculation of the whole Arctic sea ice extent normal, all grid boxes with sea ice concentration above 15% and north of 20° N are used in the calculation, including the polar hole (ERA-Interim and ERA5) and lake ice (ERA-Interim).

### 2.2. Data Set Description

For each of the sea ice coverage parameters (concentration, area, and extent), the climate normal data file contains the following fields:

- Climate normal of the parameter,
- Minimum, maximum, and standard deviation,
- 10th and 90th percentiles,
- First, second, and third quartile,
- Number of valid data records,
- Quality flag.

Sea ice concentration represents the area fraction of sea ice at each grid cell. The sea ice extent is the area within the 15% concentration contour while the sea ice area is the area-integrated concentration of all grid cells with the sea ice concentration values of 15% or higher over the given region. The North Pole hole region is excluded from the calculation of the sea ice area, which mostly impacts the calculation of the sea ice areas in the whole Arctic and Central Arctic regions.

For the sea ice concentration climate normals, the following additional fields are also included:

- Percentage of ice presence (sea ice concentration ≥15%),
- Regional mask.

Monthly and annual sea ice area and extent climate normals are in the ASCII format for each of 16 regions (that is, the Arctic and 15 regional divisions; see Table 1 for regional names and identifiers).

**Table 1.** Regional names and identifiers (IDs).

| Region | Region ID | Region | Region ID |
|---|---|---|---|
| Whole Arctic | Arctic | Barents Sea | BarentsSea |
| Japan Sea | JapanSea | Kara Sea | KaraSea |
| Okhotsk Sea | OkhotskSea | Laptev Sea | LaptevSea |
| Bering Sea | BeringSea | East Siberian Sea | EastSiberian |
| Hudson Bay | HudsonBay | Chukchi Sea | ChukchiSea |
| St. Lawrence | StLawrence | Beaufort Sea | BeaufortSea |
| Newfoundland Bay | NewfoundlandBay | Canadian Archipelago | CanadianArchipelago |
| Greenland Sea | GreenlandSea | Central Arctic Ocean | CentralArctic |

Monthly and annual sea ice concentration climate normals, on the other hand, are in the Network Common Data Form (NetCDF) format which is self-describing and machine-readable. The region divisions are denoted by the regional mask field (Figure 1).

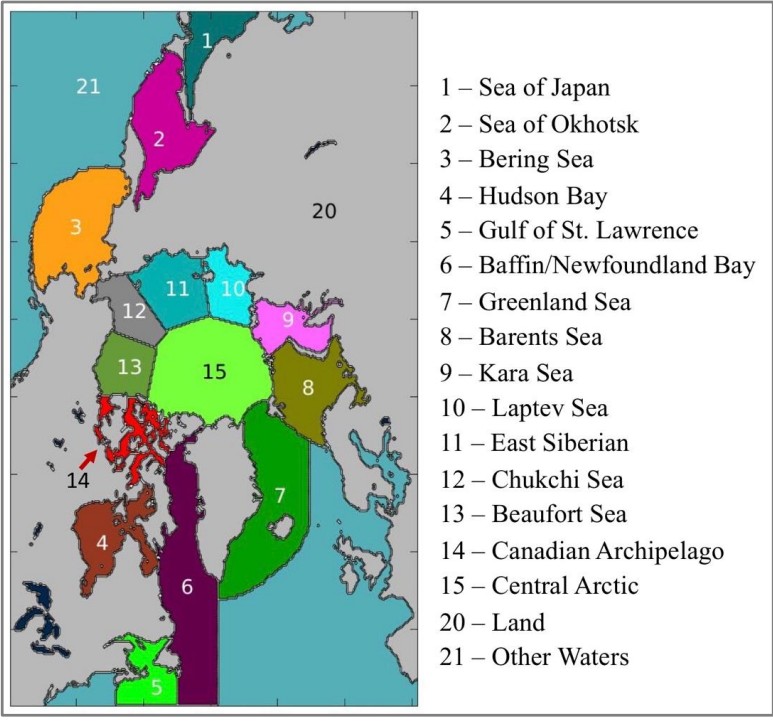

**Figure 1.** Location map of the regions in Arctic. From [3].

## 3. Methods, Variability, and Uncertainty Estimates

### 3.1. Approach to Computing Normals

The method of using the arithmetic average over the last three complete decades as defined by the World Meteorological Organization [9] is adapted. The sea ice normals are derived from the monthly $GSFC_m$ SIC values which are the averages of daily $GSFC_m$ SIC values. For the period of 1981–2010, the only period of missing monthly $GSFC_m$ SIC data is December 1987–January 1988, due to missing sensor data. Therefore, there is no time period with 3 consecutive missing SIC records.

To help users better understand spatial and temporal characteristics of the normals and input data, the spatial distribution and variability of annual SIC normal are described in Section 3.2. Spatial modes of variability of monthly SICs used for computing annual SIC normal are depicted utilizing the Empirical Orthogonal Function (EOF) analysis and described in Section 3.3. The temporal distribution of monthly Arctic SIEs for computing the monthly and annual Arctic SIE normals is captured in Section 3.4. The decadal trends of the annual Arctic SIE minimum and maximum are also described. The data uncertainty estimates and normal quality flags are presented in Section 3.5 and product accuracy in Section 3.6.

### 3.2. Spatial Distribution of Climate Normal of the Annual Sea Ice Concentration

Spatial distributions of the annual SIC climate normal and its standard deviation (STD) are shown in Figure 2a,b. High SIC values are fairly persistent in the central Arctic ocean, which resulted in overall low STD in the region. The differences of the valid data point in the North Pole hole region result from the different pole hole mask sizes (Figure 2c, Table 2).

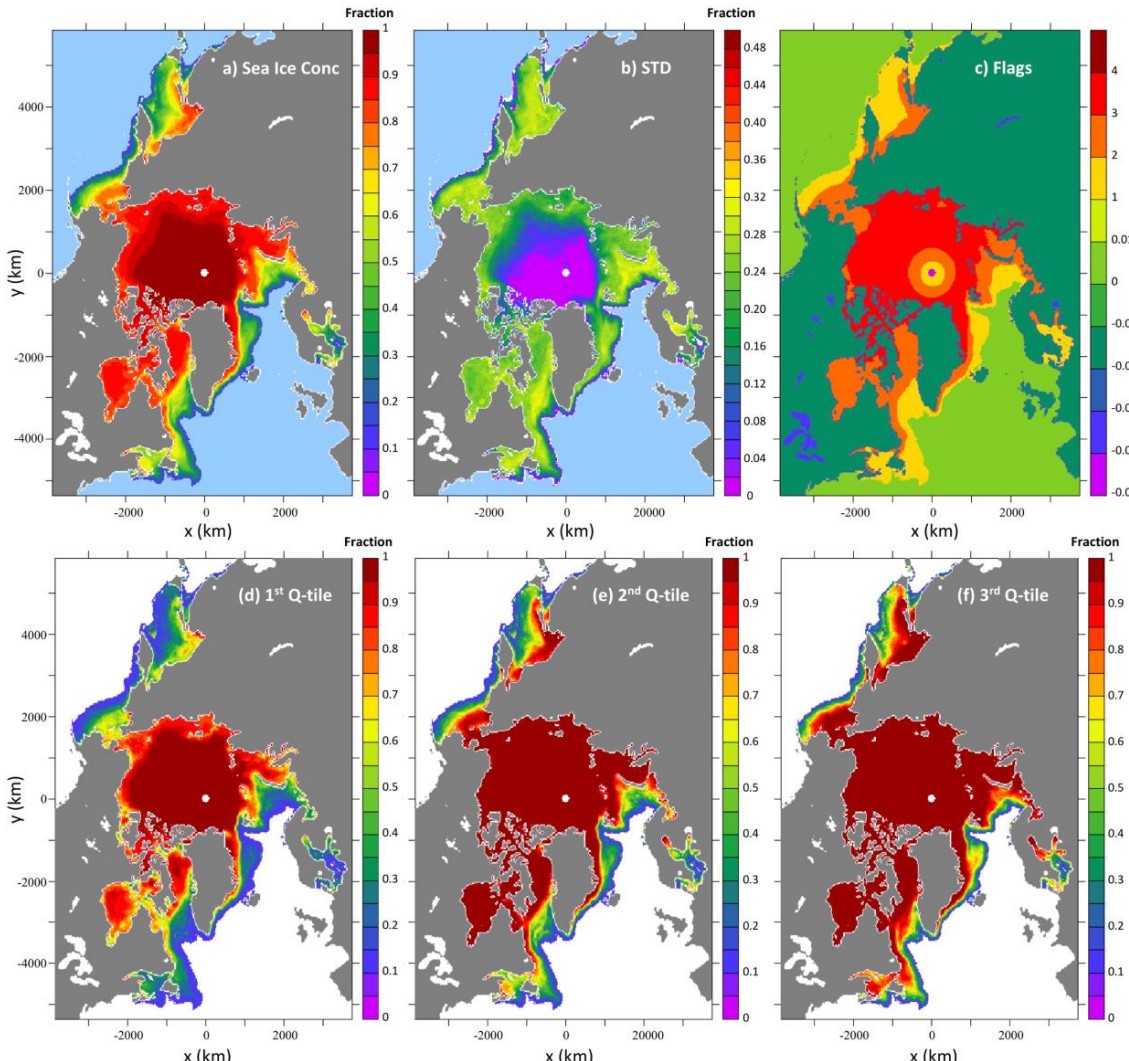

**Figure 2.** Spatial distributions of annual sea ice concentration (SIC): (**a**) climate normal, i.e., average of monthly fields for the period of 1981–2010, (**b**) standard deviation, (**c**) quality flags, (**d**) first quartile, (**e**) second quartile, and (**f**) third quartile. Light blue in (**a**,**b**) denotes water, gray denotes land, and white is for lakes, coastal, and the North Pole hole areas. The flag values of (−0.05, −0.04, −0.03, −0.02, −0.01, 0, 1, 2, 3, 4) in (**c**)) denote (the North Pole hole, Lakes, Coastal, Land, Missing data, All water, Low record, Provisional, Standard, and Complete). See Table 3 for more information.

**Table 2.** The North Pole hole attributes and the record periods used during the period of January 1981–December 2010.

| North Pole Hole Mask | North Pole Hole Area ($10^6$ km$^2$) | North Pole Hole Radius (km) | Latitude (°N) | Total Number of Grid Cells | Record Period Used |
|---|---|---|---|---|---|
| SSMIS | 0.029 | 94 | 89.18 | 44 | January 2008–December 2010 |
| SMM/I | 0.31 | 311 | 87.2 | 468 | August 1987–December 2007 |
| SMMR | 1.19 | 611 | 84.5 | 1799 | January 1981–July 1987 |

### 3.3. Empirical Orthogonal Function (EOF) Analysis

The EOF analysis has been commonly used in climate studies to examine possible spatial modes of variability and how the corresponding principal component time series change over time [18]. EOF analysis has thus been utilized to show the spatial modes of September sea ice concentration variability. In the present investigation, EOF analysis is only performed for grid cells for which at least 10 sea ice concentration values are less than 100% and for which at least 10 values are greater

than or equal to 15%. This is because EOF analysis would not be meaningful for points in which there is not sufficient variability across the September series. Thus, we exclude areas where the sea ice concentration is usually/always 100% as well as areas where the sea ice concentration values usually/always indicate ice-free conditions (less than 15% concentration). For September Arctic sea ice concentrations, the first three EOF modes account for 45% of the total variance. The first EOF mode of sea ice concentration shows a distinct spatial pattern with a solid downward trend of 9% per decade, which is significant at the 95% confidence level (Figure 3a,d) and suggests that Mode 1 largely represents a climate change signal. Modes 2 and 3 account for 13% and 11% of the total variance, respectively, and feature less-energetic cross-basin patterns (Figure 3b,c), while the associated time series do not feature a significant trend (Figure 3e,f), suggesting that these modes are likely attributable to natural variability.

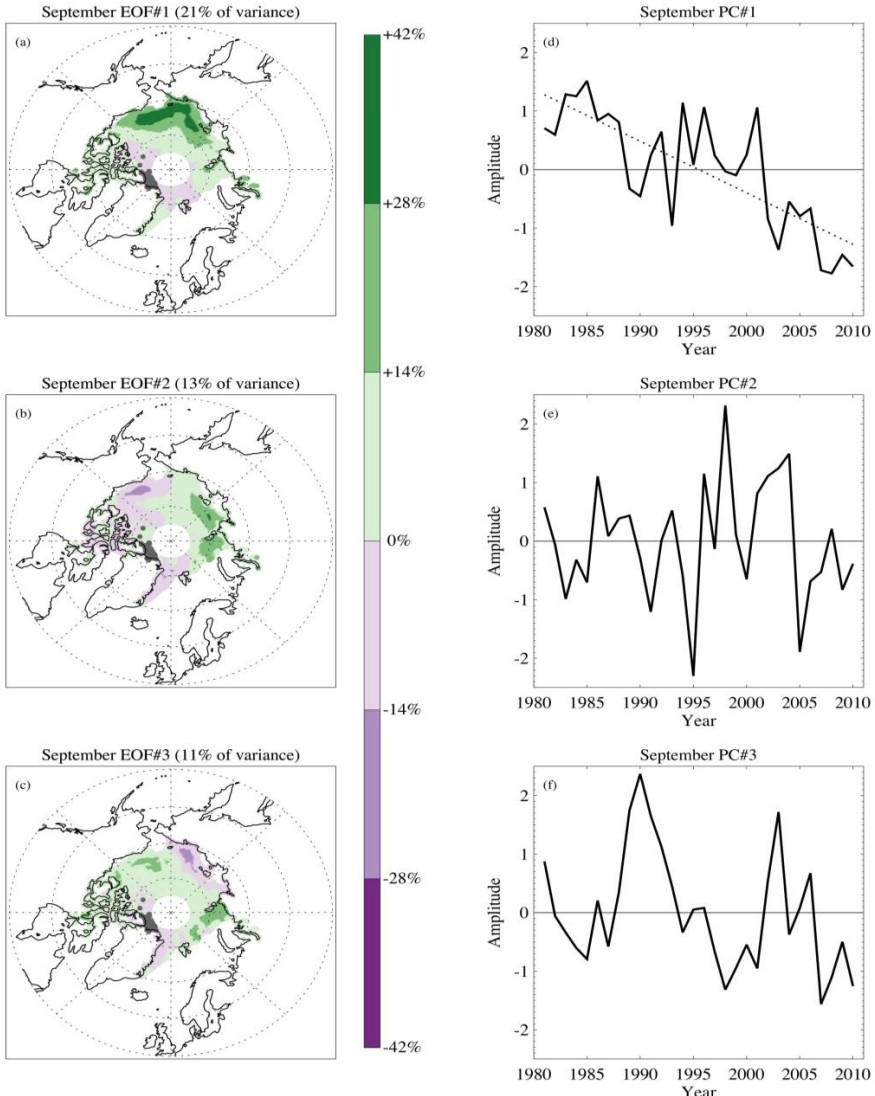

**Figure 3.** (**a**–**c**) are the spatial patterns of the first three leading empirical orthogonal functions (EOFs) for September SIC, and (**d**–**f**) are their corresponding principal component time series. The dashed line in (**d**) is the linear regression trend line. Green (purple) areas project positively (negatively) onto the associated time series, with the EOF magnitude modulating the intensity of the effect of the same (opposite) sign of the time series values.

### 3.4. Temporal Distributions and Trends of Arctic Sea Ice Extent

The distinct interannual variability of Arctic SIE and reduction in both annual maximum and minimum can be easily seen in the temporal distribution (Figure 4a). The seasonal cycle of 30-year average of monthly Arctic SIE is shown in Figure 4b with a mean of 12.19 ($10^6$ km$^2$) (Figure 4b).

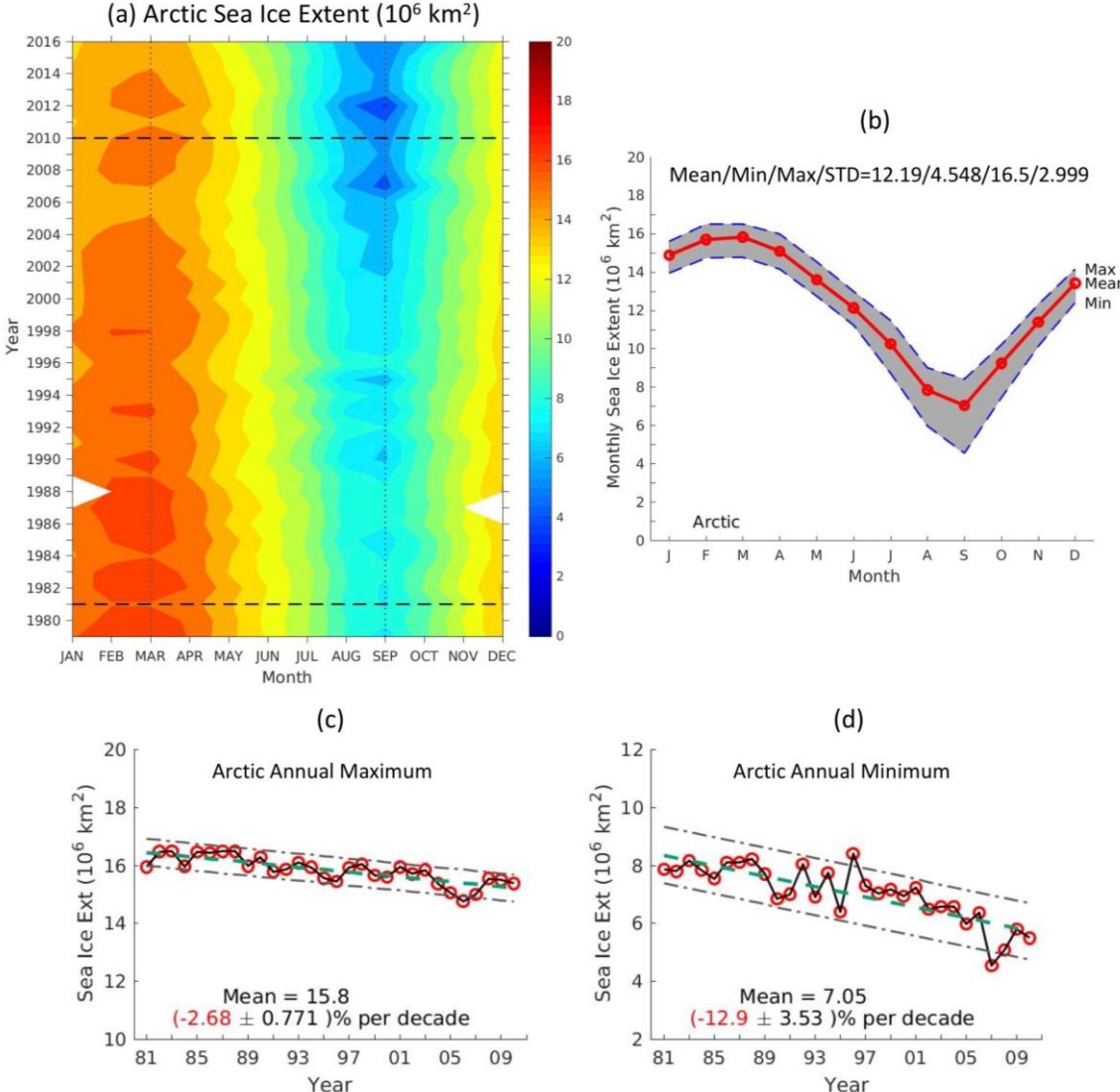

**Figure 4.** (**a**) The temporal distribution of monthly Arctic sea ice extent (SIE, $10^6$ km$^2$) for the period of 1979–2016. The horizontal dashed lines denote the beginning and the ending of the climate normal period (1981–2010). The vertical dotted lines denote the months of March and September, which are normally the annual maximum and minimum, respectively. The white space denotes missing sensor data in December 1987 and January 1988. (**b**) Seasonal cycle of 30-year average monthly SIE ($10^6$ km$^2$, thick red line with filled circles) and the maximum and minimum values for each month (dashed blue lines) over the climate normal period for the Arctic region. The lower two panels are time series of sea ice extent (red circles with solid black line), its linear regression (thick green dashed line) for the annual maximum (**c**) and minimum (**d**). The values in red are decadal trends that are significant at the 99% confidence level.

For the Arctic region as a whole and for the period of 1981–2010, the annual SIE maximum and minimum experienced significant downward trends at −2.68% and −12.9%, respectively (Figure 4c,d). These values are similar to those computed by [3] for the period of 1979–2015, which are −2.41% and

−13.5% per decade, respectively. The differences primarily reflect the removal of relatively high values for the first two years for the annual SIE maximum and the accelerated annual SIE depletion in the recent years, especially with the record low in 2012, for the annual SIE minimum.

### 3.5. Data Uncertainty Estimates and Quality Flags

For a given time frequency, that is, monthly or annual, several ways are used to represent data uncertainty estimates. The historical ranges of ice parameters are represented by their maximum and minimum values. The amount of dispersion of the values is captured by the standard derivation. The 10th and 90th percentiles as well as the first, second, and third quartile values are used to represent the frequency distribution of all values. The second quartile represents the median value of all input data, which may or may not be equal to the mean. An example is given for the annual SIE climate normal (Figure 5), which shows that the SIEs are skewed towards higher values. The spatial distributions of the quartiles of the annual sea ice concentration normal are shown in Figure 2d–f. These quartile distributions imply that, in the coastal areas of the Kara, Laptev, East Siberian, Chukchi, and Beaufort seas, the areas adjacent to the central Arctic ocean are more likely to have a higher SIC value than a lower one throughout the 30-year period, demonstrated by the fact that the values of the first quartile in those regions are close to or even slightly higher than that of the mean (comparing Figure 2d to Figure 2a).

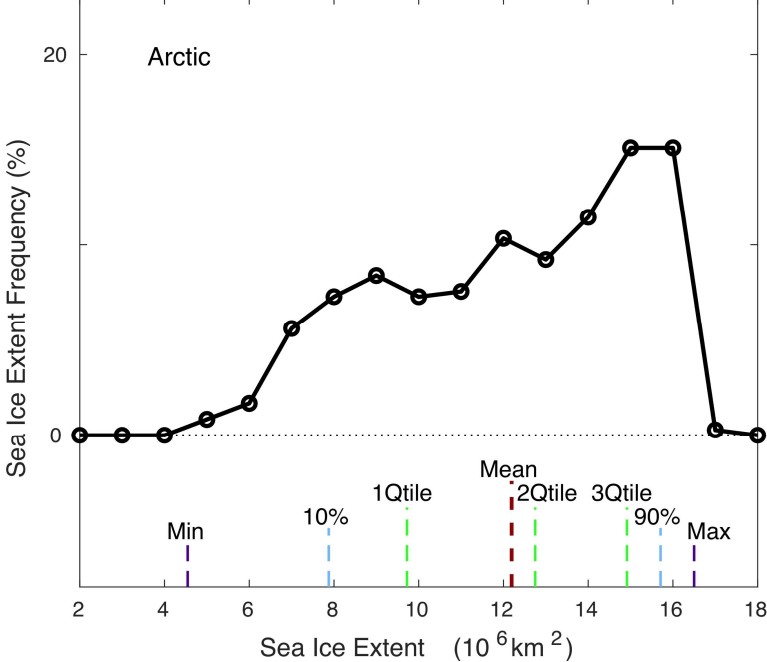

**Figure 5.** The relative frequency distribution (thick solid black line with circles) of all valid monthly SIE values going into the calculation of the Arctic annual SIE climate normal. The number of SIEs within each bin is normalized by the total number of all valid SIEs. The minimum and maximum SIE values of all valid data are denoted by the intersection points between the *x*-axis and the dashed purple vertical lines. The 10th and 90th percentiles are denoted by that of the dashed blue lines. The first, second, and third quartiles are denoted by that of the dashed green lines. The mean, which is the annual SIE climate normal, is denoted by that of the dashed red line.

Based on guidance by WMO [9], the quality of the sea ice normals has been categorized as one of the following characterizations: Low record, Provisional, Standard, and Complete. They are assigned numerical values of 1 to 4, corresponding to the valid data record number, i.e., $N_{ice}$, in years: $N_{ice} < 10$, $10 \le N_{ice} < 25$ without 3 consecutive missing years, $25 \le N_{ice} < 30$ without 3 consecutive missing years, and $N_{ice} = 30$, respectively (Table 3). The cell with SIC = 0 throughout the whole period of record is

considered to be an "All Water" cell. Quality flag names and values in the sea ice concentration climate normals are captured in Table 3. The same flag values in the CDR data for the North Pole Hole, Lakes, Coastal, and Land are adopted, which are not applicable for the sea ice area and extent normals.

**Table 3.** Flag names, conditions, and values for the sea ice concentration climate normal.

| Flag Name | Pole Hole | Lakes | Coastal | Land | Missing Data |
|---|---|---|---|---|---|
| Value | −0.05 | −0.04 | −0.03 | −0.02 | −0.01 |
| Flag Name | All Water | Low Record | Provisional | Standard | Complete |
| Condition | SIC = 0 | $N_{ice} < 10$ | $10 \le N_{ice} < 25$ | $25 \le N_{ice} < 30$ | $N_{ice} = 30$ |
| Value | 0 | 1 | 2 | 3 | 4 |

*3.6. Data Product Accuracy*

Data product accuracy is examined by comparing the Arctic monthly SIE climate normals with other data products. This approach is similar to that used by [19], who estimated an absolute extent uncertainty based on the spread of the extents computed from six different products. Figure 6 shows the time series of the Arctic monthly SIE climate normals (thick blue line with circles) and corresponding range of all the $GSFC_m$ SIC values for each month (light grey shade). Superimposed are monthly SIE climate normal values from SII (green solid line), ERA-Interim (purple dashed line), and ERA5 (red dashed line). Both SII and ERA5 SIE values are systematically lower than those computed from the $GSFC_m$ SIC values, with a difference of 0.55 and 0.5 ($10^6$ km$^2$), respectively. A difference of 0.285 ($10^6$ km$^2$) was found between $GSFC_m$ and ERA-Interim SIEs, largely due to lower ERA-Interim SIE values during the summer months. The differences can be attributed to both different numbers and locations of grid cells and different concentration values at co-located grid cells. The slightly higher ERA-Interim SIE values during the winter months are mainly the consequence of including lake ice.

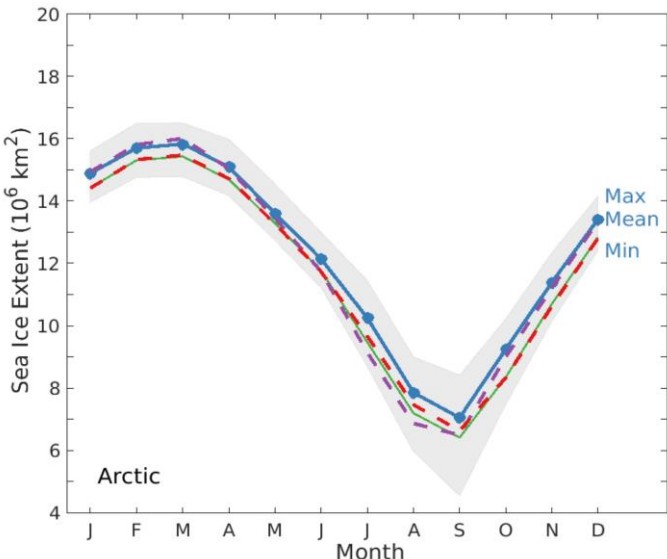

**Figure 6.** The seasonal cycle of monthly Arctic $GSFC_m$ SIE climate normal values ($10^6$ km$^2$, thick blue line with circles) bounded by the maximum and minimum values for each month over the climate normal period (1981–2010) (light grey shade), superimposed with the SIE climate normal values from SII (solid green line), ERA-Interim (dashed purple line) and ERA5 (dashed red line).

The SIE differences between $GSFC_m$ and the other products may be attributed to two main factors. One is the difference in how SIE was computed. The $GSFC_m$ and ERA-Interim SIEs were computed from the monthly SIC fields while the SII and ERA5 SIEs were the monthly average of daily SIE values.

Another is the difference in the number of cells with SIC values above the 15% SIC threshold. To help quantify the impact of how SIE was calculated, we have computed the monthly SIE values from the monthly GSFC-derived NT SIC fields included in the CDR data files. Figure 7 shows that while the overall difference between these two products is on the order of 0.55 ($10^6$ km$^2$), the impact of using the daily versus monthly SIC fields for computing the SIE values is on the order of 0.38 ($10^6$ km$^2$). Using the ERA-Interim data, the impact of using the daily versus monthly SIC fields for computing SIE values was found to be on the order of 0.29 ($10^6$ km$^2$), while that of lake ice on the order of 0.17 ($10^6$ km$^2$) using the ERA5 data (not shown). Thus, for the Arctic as a whole with a mean SIE value of 12.19 ($10^6$ km$^2$), the differences between GSFC$_m$ and other SIEs are in the range of 2.3–4.5%.

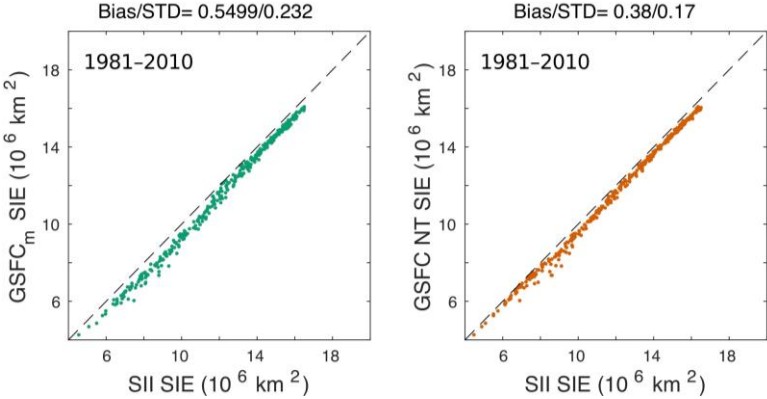

**Figure 7.** The scatter diagram between monthly SII SIEs and GSFC$_m$ SIEs (left) and GSFC-derived NT SIEs (right), respectively.

## 4. User Notes

Zero values in SIA and SIE climate normals do not necessarily mean that there is no ice in the region. They actually represent a state that the sea ice concentrations of all the grid cells within the region are less than 15%.

**Author Contributions:** G.P. designed and produced the sea ice climate normal products. A.A. and W.N.M. contributed to the early discussions on representing data uncertainty in the climate normal products. A.A. provided the guidance on the WMO climate normal calculation. He also carried out the EOF analysis of sea ice concentration climate normal and generated Figure 3. F.V. computed the monthly Copernicus sea ice coverage climate normal values derived from the daily and monthly ERA-Interim as well as that from the daily ERA5 sea ice concentration data with/out lake ice. J.C. reviewed the design for the sea ice climate normal products. P.J. provided archival guidance and helped with facilitating with the NCEI archival process, reviewing and providing feedback on file naming convention. G.P. carried out the rest of analyses and generated all other figures and tables. She also drafted the manuscript with contributions from A.A. All authors reviewed, contributed to, and approved the final version of the manuscript.

**Funding:** This research received no external funding.

**Acknowledgments:** Ge Peng is supported by NOAA's National Centers for Environmental Information (NCEI) through the Cooperative Institute for Climate and Satellites—North Carolina (CICS-NC) under Cooperative Agreement NA14NES432003. Comments from Imke Durre and an NCEI internal reviewer are beneficial in improving the clarity of the paper. We thank Scott Stevens for proofreading the manuscript. Constructed comments and suggestions from two anonymous reviewers are beneficial for improving the clarity of the paper. The authors wish to acknowledge use of the Ferret program for analysis in calculating sea ice climate normals. Ferret is a product of NOAA's Pacific Marine Environmental Laboratory. (Information is available at http://ferret.pmel.noaa.gov/Ferret/). ECMWF implements the Copernicus Climate Change Service (C3S) on behalf of the European Commission.

**Conflicts of Interest:** The authors declare no conflict of interest. The funders had no role in the design of the study; in the collection, analyses, or interpretation of data; in the writing of the manuscript, or in the decision to publish the results.

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
