# Peer review of "Sea Ice Climate Normals for Seasonal Ice Monitoring of Arctic and Sub-Regions"

_data, 2019_

Round 1

Reviewer 1 Report

This manuscript (Data Descriptor) and its output products are very useful for the Arctic sea ice research community and as a result should be published after minor corrections.

Overall, the manuscript is written clearly and succinctly with appropriate steering of the reader to published work along similar lines when appropriate. 

I have two main concerns with the manuscript. 

1) Section 3.3 on EOF is too brief as it currently reads. The authors should add more description of what PC1 means from a spatial perspective on their plots. In other words, please add some basic description as to what the green and purple regions mean on their plots. Furthermore, add the % of PC1, PC2, etc onto each of the plots (summing to 45%). Often, in EOF, PC1 accounts for the vast majority of variability while PC3 can account for very little. These values provide additional important information to the experienced reader.

2) I don't see the necessity of having section 4 - User Notes as a separate section unless this is a requirement of the journal (Data Descriptor) format style. Surely the material in it currently can be added to the discussion of 15% SIC within lines 90-92 within the Data Description section. Is it a separate section because the authors envision subsequent editions in future years where user information from subsequent feedback can be added?

Minor points  

1) Pls refrain from using the term 'spread' (in both the abstract and elsewhere). Use the term range. Range is a statistical term and is appropriate in this context. In fact, the discussion in lines 175-177 can be removed since this description of what 'spread' is meaning is not needed.

2) 'data points' is not really necessary in Figure 2 caption. I suggest removing.

3) In Figure 2 caption ... the -0.01 flag value is missing (representing Missing Data .. which I deduced from Table 3)

4) Question/Comment ... in Figure 2b (STD) .. why does the outermost boundary of sea ice in the MIZ have a std value which is lower than the ice to the north of it (ie. blue boundary) ... or am I missing something here? In other words, the Arctic SIC standard deviations progress from the lowest nearest to the north pole and progressively increases towards the south ... as it should ... but then increases at the outermost boundary? Please explain? .. or is this a plotting typo?

5) Figure 3. Why is the EOF analysis restricted to just north of 60oN? ... when the spatial domain (Figure 1) includes regions much further south?

6) Line 213 ... remove "It is easy to see that the" ... it is not needed.

7) The font types and sizes are quite different between the various figures and tables. For the final version it would be a professionally touch to have them all as similar as possible. 

Reviewer 2 Report

Peng et al. presented the sea ice climate normal from 1981 to 2010. The topic of this study is interesting for Arctic research community regarding to the spatial and temporal characteristics of sea ice in the last few decades, and the differences of multiple sea ice data sets. I found this manuscript is well written and clearly presented for most of the part.

My comments are:

Line 69-71 is really confusing. Which data set did you use? the CDR or the GSFC sea ice concentration field. This needs clarification. What does manually quality-control mean?

Line 75-77, is this the same NT algorithm as those in previous two paragraphs?

Line 78-80, do ERA-I and ERA-5 assimilate any passive microwave data or other sea ice concentration data? If they do, what are those data? 

Line 125, can you explain what is the purpose of showing figure 2d, 2e, and 2f, except the comparison of 2d and 2e in a later paragraph?

Line 145, can you briefly explain what these EOF patterns represent, and what the trend mean?

Line 182, I am wondering the purpose of showing figure 5 without any explanation? And Figure 5 is really confusing and hard to understand.

Line 183, it is really confusing what "they" refers to. You might want to rewrite it.

Table 3, it is not very common to use float number as flag values.

Line 224-226, "To help quantify...", I do not see how you can quantify the impact in the way described in this sentence.

Round 2

Reviewer 1 Report

The authors have done an admirable job at revising their manuscript based on the two reviews. I am satisfied that this manuscript is now ready for publication.

Reviewer 2 Report

The authors addressed all my comments.